# Combined Therapy Using Human Corneal Stromal Stem Cells and Quiescent Keratocytes to Prevent Corneal Scarring after Injury

**DOI:** 10.3390/ijms23136980

**Published:** 2022-06-23

**Authors:** Vishal Jhanji, Mithun Santra, Andri K. Riau, Moira L. Geary, Tianbing Yang, Elizabeth Rubin, Nur Zahirah Binte M. Yusoff, Deepinder K. Dhaliwal, Jodhbir S. Mehta, Gary Hin-Fai Yam

**Affiliations:** 1Corneal Regeneration Laboratory, Department of Ophthalmology, University of Pittsburgh, Pittsburgh, PA 15213, USA; jhanjiv@pitt.edu (V.J.); mithun.santra@pitt.edu (M.S.); mlo39@pitt.edu (M.L.G.); tianbing@pitt.edu (T.Y.); eer35@pitt.edu (E.R.); dhaliwaldk@upmc.edu (D.K.D.); 2Tissue Engineering and Cell Therapy Group, Singapore Eye Research Institute, Singapore 169856, Singapore; andri.kartasasmita.riau@seri.com.sg (A.K.R.); zahirahmy@gmail.com (N.Z.B.M.Y.); jodmehta@gmail.com (J.S.M.); 3Ophthalmology and Visual Sciences Academic Clinical Program, Duke-NUS Medical School, Singapore 169857, Singapore; 4McGowan Institute for Regenerative Medicine, University of Pittsburgh, Pittsburgh, PA 15213, USA

**Keywords:** corneal scarring, cell therapy, corneal stromal stem cells, corneal stromal keratocytes, mouse corneal injury

## Abstract

Corneal blindness due to scarring is conventionally treated by corneal transplantation, but the shortage of donor materials has been a major issue affecting the global success of treatment. Pre-clinical and clinical studies have shown that cell-based therapies using either corneal stromal stem cells (CSSC) or corneal stromal keratocytes (CSK) suppress corneal scarring at lower levels. Further treatments or strategies are required to improve the treatment efficacy. This study examined a combined cell-based treatment using CSSC and CSK in a mouse model of anterior stromal injury. We hypothesize that the immuno-regulatory nature of CSSC is effective to control tissue inflammation and delay the onset of fibrosis, and a subsequent intrastromal CSK treatment deposited collagens and stromal specific proteoglycans to recover a native stromal matrix. Using optimized cell doses, our results showed that the effect of CSSC treatment for suppressing corneal opacities was augmented by an additional intrastromal CSK injection, resulting in better corneal clarity. These in vivo effects were substantiated by a further downregulated expression of stromal fibrosis genes and the restoration of stromal fibrillar organization and regularity. Hence, a combined treatment of CSSC and CSK could achieve a higher clinical efficacy and restore corneal transparency, when compared to a single CSSC treatment.

## 1. Introduction

A transparent cornea is essential for normal vision. Typically, the human cornea has 5 layers, starting with the outermost corneal epithelium, followed by the Bowman’s layer, corneal stroma, Descemet’s membrane, and the innermost corneal endothelium. The corneal stroma occupies about 80 to 90% of the overall corneal volume. It contains a highly organized stromal matrix, with regularly aligned collagen fibrils organized in a form of lamellae which run orthogonally throughout the stroma [1,2]. Corneal stromal keratocytes (CSK), the major type of stromal cells, synthesize and deposit collagens and keratan sulfate proteoglycans (KSPGs; lumican, keratocan, and mimecan) to regulate stromal matrix organization, contributing to the mechanical strength and optical clarity of the cornea. CSK remain quiescent at the G0 stage of the cell cycle throughout adult life; however, after stromal injury or diseases, the surviving CSK in the affected region activate and transit to the proliferative stromal fibroblasts (SF), which produce repair-type extracellular matrix (ECM) proteins (including fibronectin, proteinases, and integrins) and engage in wound healing events [3,4,5]. SF further transform into highly contractile myofibroblasts, which produce and deposit excessive ECM proteins, including collagens and fibronectin, and reduce stromal crystallin expression. This myofibroblast function not only serves to close the wound, but also forms scar tissues in the transparent cornea, leading to corneal opacities which can block the passage of light and impair vision [2,6].

Worldwide, corneal opacification and scarring is a significant cause of global blindness. The standard treatment of corneal scarring is allogenic donor corneal tissue grafting to replace the scarred corneal layers (either in the form of penetrating or lamellar keratoplasty). Although corneal transplantation has a remarkable degree of success, there are various limitations, the chief being it relies on the availability of transplantable donor tissues, which is severely limited in many countries [7]. In global terms, it is estimated that only one in 70 patients with corneal scarring has access to donor corneal tissue [8]. Thus, there has been an increasing interest to develop therapeutic alternatives, including cell-based and cell-free strategies, bioengineered constructs and scaffolds, as well as corneal protheses. However, no effective and long-lasting solutions have been reported [9,10,11,12].

Cell-based therapies have been reported to successfully prevent corneal scarring in animal models of corneal wound injury [13,14]. The treatment using stromal cells expressing crystallins, collagens, and KSPGs has been shown to restore native stromal architecture and improve corneal transparency. The intrastromal injection of CSK reduced corneal haze in a rat model of corneal injury induced by laser photo-ablation [14,15]. Stromal injection or topical application of human corneal stromal stem cells (CSSC) has been shown to block corneal tissue scarring in mice with anterior stromal injury [13,16,17,18]. CSSC are located in the anterior limbal stroma and are the mesenchymal stem cell type of progenitors that can differentiate into keratocytes. They were reported to control corneal tissue inflammation and fibrosis [17,18,19]. Our recent study has identified that 2 specific miRNAs (miR-29a and 381-5p), delivered via CSSC-derived EV, could downregulate stromal inflammation and fibrotic onset after corneal injury in mice, leading to a scarless stromal ECM remodeling [20].

The use of a functional cell type is a pre-requisite for cell therapy. To obtain CSSC with good healing potency has been challenging, and batch-to-batch variation exists from different donors. In addition, lack of screening standards for donor tissues and cultivated CSSC make it less feasible to determine which cell batches are optimal to effectively remodel the corneal scar. Therefore, it has been observed that even though CSSC treatment reduces corneal scarring, additional strategies are required to improve corneal clarity. In this study, we examined a combined cell-based treatment with CSSC and CSK (both cell types originated from the same donor cornea) to reduce corneal scarring in a mouse model of anterior stromal injury. We hypothesize that the immuno-regulatory nature of CSSC is effective to stabilize the wound by controlling tissue inflammation and regulating the immune response, thereby delaying the onset of fibrosis. A subsequent intrastromal CSK injection could deposit stromal-specific collagens and KSPG to improve the stromal matrix remodeling.

## 2. Materials and Methods

### 2.1. Donor Corneas, Stromal Cell Isolation, and Culture

The research followed the tenets of the Declaration of Helsinki and was approved by the University of Pittsburgh Institutional Review Board (IRB) and the Committee for Oversight of Research and Clinical Training Involving Decedents (CORID), Protocol #161. Human corneas, approved for research purposes, from de-identified donors (information in Appendix A), were obtained from the Center for Organ Recovery and Education, Pittsburgh, PA, USA (www.core.org, accessed on 24 May 2022). Tissues were preserved in Optisol GS (Bausch & Lomb, Rochester, NY, SUA) and used within 9 days post-enucleation. After the clearing of the corneal epithelium and endothelium by gentle scraping and rinses, the central stroma (8 mm diameter) was isolated and trimmed into small pieces for digestion with 0.1% collagenase (NB 6 GMP grade, Nordmark Pharm GmbH, Uetersen, Germany) and 0.1% AlbuMAX^TM^ I (Thermo Fisher Sci., Waltham, MA, USA) for 8 h at 37 °C (see the schematic diagram in Figure 1A). After passing through a cell strainer (70 μm pore size, Corning, NY, USA), the single cell suspension was seeded on a collagen I-coated culture surface and propagated using ERI reagents added with 0.5% heat-activated fetal bovine serum (FBS, Gibco) until passage 4 [21]. The culture was switched to a serum-free ERI condition for 1 week to generate quiescent CSK (q-CSK). After characterization, q-CSK expressing keratocan, ALDH3A1, and lumican were used for intrastromal injection, similar to our earlier study [14]. In contrast, the corneal rim was used for human CSSC isolation. The anterior limbal stroma (0.5 to 1 mm wide, 0.2 mm deep) was carefully isolated and digested with 0.1% collagenase for 10 h, similar to the process used above (see the schematic diagram in Figure 1A). After passing through a cell strainer, the single cell suspension was seeded on an FNC-coasted culture surface, and the cells were propagated with stem cell growth medium (JM-H) containing 2% (*v*/*v*) pooled human serum (Innovative Res., Novi, MI, USA) [13,22]. Cells with clonal growth were expanded to passage 2 (P2) for characterization and passage 3 for animal experiments. CSSC expressing markers for MSC (CD73), pluripotency (ABCG2 and nestin), and stromal cells (ALDH3A1) were used for corneal wound treatment. To generate stromal fibroblasts (SF), human CSK at passage 2 were changed to DMEM/F12 medium (Gibco) supplemented with 10% FBS for 2 passages to ensure a complete transition. SF were characterized to express fibronectin and collagen III, as described earlier [23].

### 2.2. Medium Formulations

JM-H for CSSC culture: DMEM (1 g/L D-glucose, Gibco 10567-014), MCDB 201 (Sigma-Aldrich M6770), added with insulin-transferrin-selenite (ITS, 0.5×, Gibco 41400-045), AlbuMAX I (1 mg/mL, Gibco 11020-021), L-ascorbate-2-phosphate (0.5 mM, Sigma-Aldrich A8960), recombinant human EGF (10 ng/mL, Sigma-Aldrich E9644), recombinant human PDGF-BB (10 ng/mL, R&D 520-BB), dexamethasone (10 nM, Sigma-Aldrich D4902), penicillin/streptomycin (BioWhittaker, Lonza, Walkersville, MD, USA), and 2% pooled human serum (Innovative Res.).

ERI for CSK culture: DMEM/F12 (Gibco 10565-018), MEM amino acids (Gibco 11130-051), MEM non-essential amino acids (Gibco 11140-050), ITS (0.5×), AlbuMAX I (1 mg/mL), L-ascorbate-2-phosphate (0.5 mM), recombinant insulin growth factor 1 (10 ng/mL, Sigma-Aldrich I3769), Y27632 (10 nM, Chemdea CD0141), human amnion stromal extract (5 μg protein/mL, preparation following the method of Yusoff et al. [24], and penicillin/streptomycin.

### 2.3. Cell Characterization

**Cell viability**—The cell suspension (10 μL) was mixed with 0.4% trypan blue (10 μL) and loaded to a hemacytometer, and the number of viable (non-trypan blue stained) and non-viable cells (trypan blue stained) were quantified using a Countess cell counter (Thermo Fisher). Percentages of viable cells were compared, and *p* values were calculated using one-way ANOVA.**Growth measurement**—Real-time cell growth analysis was performed using the xCELLigence system RTCA SP (Agilent, Santa Clara, CA). Cells (5 × 10^3^ cells) were grown in an E-plate 96, with gold electrodes at the bottom of each well. The growth assay was performed according to the manufacturer’s instructions. Cell indices at the start of the log phase were normalized, and the doubling time was calculated by RTCA Software Pro (Agilent).**Spheroid forming assay**—Cells were plated at a density of 200 cells per well of an Ultra-Low Attachment 6-well plate (Corning Coster) in a spheroid medium with Advanced DMEM containing B27 (1:50, Gibco), basic FGF (10 ng/mL, Gibco), EGF (10 ng/mL, Gibco), and antibiotics. At day 7, the percentage of spheroid formation was quantified.**Collagen I and III production**—Culture supernatants were collected and spun to remove cell debris. The secretion of pro-collagen I was assessed with Human Pro-Collagen I α1 (Pro-COL1a1) DuoSet ELISA (R&D Systems, Minneapolis, MN), and collagen III with Human Collagen, type III, α1 (COL3a1) ELISA kit (Cusabio, Houston, TX, USA) according to manufacturer’s protocol.**TSG-6 expression**—The cells were seeded at a density of 5000 cells per well of a 24-well plate overnight. They were then treated with TNFα (20 ng/mL) for 24 to 72 h. Total RNA was collected from RLT lysates, and qPCR was performed to examine *TSG-6* expression (for primer information, see Appendix A).**Anti-inflammatory assay** by suppressing mouse macrophage induction to osteoclast formation—RAW264.7 cells (American Type Cell Collection, Manassas, VA) in DMEM with 5% FBS were seeded at 2 × 10^4^ cells per well of a 24-well plate overnight. The cells were treated with RANK-L peptide (50 ng/mL, Sigma-Aldrich, St Louis, MO) and Concanavalin A (20 μg/mL, ConA, Sigma-Aldrich) in the presence of native or heat-denatured conditioned media concentrate (500 μg protein) from CSSC or q-CSK cultures. After 48 h, cells were harvested in RLT buffer (Qiagen, Hilden, Germany) for total RNA extraction, followed by qPCR for osteoclast markers: tartrate-resistant acid phosphatase (*ACP5*), matrix metalloproteinase 9 (*MMP9*), and cathepsin K (*CTSK*) (for primer information, see Appendix A). The delta Ct was determined by comparison with housekeeping 18S.

### 2.4. Preparation of Conditioned Medium Concentrates

CSSC or CSK at passage 3 were grown to 50% confluence, washed, and replenished with serum-free defined medium (DMEM/F12 containing insulin-transferrin-selenite, MEM essential and non-essential amino acids, and antibiotics). After 72 h, conditioned media (CM) was collected and spun at 500 g to remove cell debris. Clear supernatant was concentrated using a MicroCon centrifugal filter (YM-100 membrane, Millipore) at 12,000 g for 10 min at 4 °C until 1/10 of the original volume remained. The total protein content of CM concentrate (CMconc) was quantified using a Pierce BCA Protein Assay kit (Thermo Fisher Sci., Waltham, MA, USA).

### 2.5. Mouse Anterior Corneal Stromal Injury Model and Treatment with Topical CSSC or Intrastromal CSK Injection

The study was carried out in accordance with the guidelines for the Care and Use of Laboratory Animals of NIH and The Association for Research in Vision and Ophthalmology Statement for the Use of Animals in Ophthalmic and Vision Research. The protocol was approved by the Institutional Animal Care and Use Committee of the University of Pittsburgh (Protocol 18022511). The Swiss Webster mice, of both genders, at 6 to 8 weeks of age were anesthetized with an intraperitoneal ketamine (50 mg/kg) and xylazine (5 mg/kg) injection. Right eyes received topical proparacaine hydrochloride (0.5%, Alcaine^®^, Alcon, Fort Worth, TX, USA) for local analgesia. After saline rinses, the central corneal epithelium (2 mm diameter, sparing the limbus) was removed using the high-speed rotation of the AlgerBrush II (Accutome Inc., Malvern, PA, USA) and scraped with a surgical blade #15 [17]. The basement membrane and anterior stroma were damaged by a second burr with the AlgerBrush. After rinses with normal saline and briefly drying with a sterile cotton spear, the wounded stromal bed was overlaid with 1 μL fibrinogen (20 mg/mL, Sigma-Aldrich) containing CSSC (30, 50, and 70 × 10^3^ cells, respectively; each group had 6 corneas), followed by 0.5 μL thrombin (100 U/mL, Sigma-Aldrich). The fibrin gel was formed within 1 min (schematic diagram in Figure 1B). Controls were the fibrin-only treatment and the non-treated injured corneas. The eyes were treated with topical tobramycin ophthalmic solution (0.3%, USP, Somerset Therapeutics, Hollywood, FL) daily. Intrastromal quiescent CSK (q-CSK) injection was performed on wounded corneas at 1-week post-Algerbrush injury or topical CSSC treatment until corneal epithelium healed. An anterior stromal tunnel was created at the corneal periphery with a 31-gauge (G) needle [14]. A volume of 0.5 μL CSK suspension (containing 10, 20, and 30 × 10^3^ q-CSK, respectively; each group had 6 corneas) in sterile saline was injected through a 33-G needle attached to a Hamilton syringe (Hamilton Co., Reno, NY, USA) (see the schematic diagram in Figure 1B). After injection, the mouse eyes received topical TobraDex (Alcon, Fort Worth, TX, USA) 2 times daily for two weeks. Ophthalmic examinations were performed on anesthetized mice weekly, and the mice were euthanized at 2 weeks post-treatment for cornea collection.

Combined CSSC and CSK treatment was performed on 15 mice receiving Algerbrush ablation to induce stromal injury on the right eyes. After saline rinses, the wounded cornea surface (n = 5 corneas) was overlaid with 0.5 μL fibrinogen containing CSSC (50 × 10^3^ cells), followed by 0.3 μL thrombin to form fibrin gel. The treated eyes received tobramycin eye drops daily. After one week, q-CSK suspension (20 × 10^3^ cells) was injected intrastromally to the CSSC-treated eyes. The eyes were then instilled with TobraDex twice daily, and the mice were euthanized at 2 weeks post-treatment for cornea collection. Controls were fibrin-only treated and saline-injected injured corneas.

The experiments were conducted using 3 pairs of CSSC and q-CSK from different donors. The corneal wounding, treatment, and cell injections were performed by a single operator (GY). Mouse corneas that received a single injection shot and with intact bleb formation covering the central corneal region were accepted for evaluation (Figure 1B) in order to maintain treatment consistency and eliminate corneal damages due to repeated needle injury.

### 2.6. Ophthalmic Examination and Measurements

At weekly intervals, mice were anesthetized using an intraperitoneal injection of ketamine and xylazine. The cross-sectional corneal structure was examined with a Spectral domain OCT (SDOCT, Bioptigen, Durham, NC, USA) with a pachymetric scan of 4 × 4 mm diameter. Scanning data were analyzed in a masked fashion. Images were processed with NIS Elements software (Nikon Inc.). The central corneal thickness (CCT) was measured as the mean of 3 measurements taken at the center (0 mm) and at 0.5 mm on either side [17]. Scar area analysis was conducted with ImageJ (National Institute of Health). Threshold images were generated after removing the corneal epithelium. Control eyes were used to set the threshold, and the percentage of scar area changes beyond the control threshold was recorded.

### 2.7. Quantitative Polymerase Chain Reaction (qPCR)

Mouse eyes with and without injuries were enucleated at 2 weeks post-injury and treatment. The isolated corneas were placed in ice-cold RLT buffer (Qiagen) with 1% β-mercaptoethanol (Sigma-Aldrich) and disrupted in MagNA Lyser Green Beads (Roche, Basel, Switzerland) at 6000 rpm for 50 s for 6 cycles in a MagNA Lyser (Roche), with intermittent cooling between cycles. The lysate was passed through a Qiashredder (Qiagen), and total RNA was extracted using RNeasy Miniprep kit (Qiagen) and an on-column RNase-free DNase kit, respectively. Reverse transcription of RNA (500 ng) was done with SuperScript III Reverse Transcriptase kit (Thermo Fisher) and random primer hexanucleotides (10 ng/mL, Thermo Fisher). Target gene expression was assayed with specific primer pairs (Appendix A) using SYBR Green Real-Time Master Mix (Life Technologies, Carlsbad, CA, USA) in a QuantStudio 3 Real-Time PCR System (Applied Biosystems). Experiments were run as technical triplicate. The relative RNA abundance was determined by ΔΔCT after normalization with the housekeeping 18S genes, and fold changes were expressed as mean ± SD. Significance was determined by a non-parametric Mann–Whitney U test.

### 2.8. Immunofluorescence

Cultured cells were fixed in 2% neutral-buffered paraformaldehyde (EMS) for 15 min on ice. Samples were treated with ice-cold 50 mM ammonium chloride (Sigma-Aldrich), saponin permeabilized, and blocked with 2% bovine serum albumin (BSA; Sigma-Aldrich) and 5% normal goat serum (NGS; Invitrogen, Carlsbad, CA, USA), followed by incubation with primary antibodies (Appendix A) for 2 h at room temperature. For keratocan and lumican staining, samples were pre-treated with endo-β-galactosidase (1.5 U in 10 mM phosphate buffer pH 7.4; Sigma-Aldrich) for 30 min at 37 °C prior to blocking and antibody incubation. After PBS washes, the signals were revealed with Red-X– or Alexa 488–conjugated IgG secondary antibody (Jackson ImmunoResearch Labs, West Grove, PA, USA) for 1 h incubation at room temperature. The samples were washed, mounted with Fluoroshield with DAPI (40,6-diamidino-2-phenylindole; Santa Cruz Biotech, Santa Cruz, CA, USA), and viewed under fluorescence microscopy (FluoView 1000 confocal microscopy, equipped with CellSens Dimension 2.1 imaging software v.2.1, Olympus, Tokyo, Japan). Alternatively, isolated mouse corneas were fixed in 2% neutral-buffered paraformaldehyde for 6 to 8 h at room temperature. After PBS washes, the samples were sucrose-infiltrated and embedded for cryo-sectioning at a thickness of 8 µm. Sections were blocked and permeabilized in 0.5% Triton X-100 (Sigma-Aldrich), 2% BSA, and 5% NGS for 60 min, followed by incubation with primary antibodies specific for corneal stroma and scarring proteins (Appendix A) for 2 h at room temperature or overnight at 4 °C. After secondary antibody labeling, the samples were processed and viewed as described above.

### 2.9. Transmission Electron Microscopy (TEM) and Morphometry

The central anterior stroma of 3 corneal samples per group were fixed sequentially with 3% glutaraldehyde (EM Sciences, Hatfield, PA, USA) in 0.1 M sodium cacodylate (pH 7.4; Sigma-Aldrich), 1% tannic acid (Sigma-Aldrich), and 1% aqueous solution of osmium tetroxide (EM Sciences) and processed for epon-araldite embedding [25]. Ultrathin sections (85–90 nm thick) were obtained with a Leica Ultracut UCT ultramicrotome equipped with a diamond knife. They were stained with 3% aqueous uranyl acetate (EM Sciences) and lead citrate. Micrographs were captured at 80 kV with a JEOL-1400 transmission electron microscope (JEOL) equipped with a Gatan Orius wide-field side mount charge-coupled digital camera (Gatan Inc.). A cross-sectional view of the stromal area (~15 random fields per sample) was captured by investigators who were masked to the experimental details. To assess the 360° fibril distribution profile, a series of concentric circles with radii increasing at 200 nm and spanning from 0 to 1600 nm range were overlaid on the stromal image and aligned to a random collagen fibril (Appendix A). The number of fibrils intersecting with each circle line along the distance from the selected fibril was quantified using the count tool of Adobe Photoshop 23.2.2. The fibril distribution profile was plotted with the number of fibrils against the distance. Alternatively, the inter-fibrillar distance between a selected collagen fibril with its surrounding unblocked fibrils was measured using the ruler tool of Photoshop (Appendix A). The mean inter-fibrillar distance of at least 5 randomly selected fibrils in 3 repeated experiments was quantified, and the data were expressed in a box plot with mean/SD and median calculation.

### 2.10. Statistics

All experiments were performed in triplicate, and the animal number was 5 or more in each group. Data were presented as mean ± SD. Mean value was compared using the unpaired two-tailed Student’s *t*-test or ANOVA, with a post hoc Bonferroni test, using GraphPad Prism 7. Non-parametric comparison was done using the Mann–Whitney U test. *p* < 0.05 was considered statistically significant.

## 3. Results

### 3.1. Human CSSC and CSK Characterization

Under phase contrast microscopy, primary CSSC at P2 underwent extensive proliferation. Holoclones containing small cells were frequently observed (Figure 2A). In contrast, primary activated CSK in a low serum culture showed convoluted cell bodies, with slender dendritic processes (Figure 2A). After serum-free conversion, the quiescent CSK (q-CSK) exhibited long and extended cell processes, with distinct elongated nuclei (Figure 2A), similar to their in vivo morphology [2]. Using the xCelligence platform, both cell types had a high viability level of (>85%) (Figure 2D; Appendix A). Since q-CSK were growth arrested in the serum-free condition, the cell doubling time could not be determined (Figure 2D). Primary CSSC replicated every 31.1 ± 1.2 h and activated CSK every 42.7 ± 3.2 h. When cultured under low-attachment conditions, CSSC generated free-floating spheres (11.9 ± 4.3 spheres per 100 cells), whereas q-CSK did not (Figure 2D).

Using immunofluorescence, q-CSK strongly expressed keratocyte markers, including keratocan, lumican, and ALDH3A1, but were devoid of Ki67 (a cell proliferation marker), whereas CSSC negligibly expressed keratocan, but were positive for Ki67, lumican, and ALDH3A1. The expression of cell-type specific markers was further screened by qPCR. As shown in Figure 2B, CSSC expressed genes known to be associated with stem cells, including *Pax6*, *ABCG2*, *nestin*, and *CXCR4*, whereas these stem cell gene expressions were suppressed in q-CSK. Alternatively, q-CSK strongly expressed keratocyte-specific genes, such as *keratocan*, *B3GnT7*, *CHST6*, *Lum*, *ALDH3A1*, *AQP1*, and *CD34*.

Collagen secretion by different stromal cell types was quantified by ELISA, using conditioned media collected after 72 h of culture. In the CM concentrates (CMconc), pro-COL1A1 was significantly secreted by the q-CSK culture, whereas CSSC and activated CSK showed a minimal release (q-CSK: 942 ± 187 pg/μg protein; CSSC: 87 ± 14 pg/μg protein, and activated CSK: 63 ± 35 pg/μg protein) (*p* < 0.05, Mann–Whitney U test) (Figure 2E). A moderate level of COL1A1 secretion was seen for stromal fibroblast (SF) cultures (572 ± 298 pg/μg protein). On the other hand, there was negligible release of COL3a1 by CSSC, as well as activated and quiescent CSK (<70 pg/μg protein), but the release was significantly produced by SF cultures (248 ± 75 pg/μg protein) (*p* < 0.05) (Figure 2E).

TSG-6 is a hyaluronan-binding protein interacting with chemokine CXCL8 to suppress neutrophil migration [26]. After treatment with TNFα, primary CSSC cultures showed upregulated *TSG-6* expression at both 24 and 72 h, when compared to the control cells without TNFα induction (24 h: 9.2 ± 2.7 folds, and 72 h: 23.8 ± 6.4 folds more than the control) (Figure 2F). On the other hand, TSG-6 stimulation was not detected in q-CSK cultures similarly treated with TNFα. Alternatively, in a chronic pro-inflammatory assay examining osteoclast differentiation of mouse RAW macrophages after treatment with RANKL and ConA, CMconc samples from CSSC dose-dependently suppressed the up-regulated expression of osteoclast gene markers (*ACP5*, *MMP9*, and *CTSK*) during RAW transition to osteoclasts (Figure 2G). However, the treatment with q-CSK CMconc did not alter the gene expression. Results of these assays indicated that CSSC possessed anti-inflammatory potency, which was not shown by q-CSK.

### 3.2. Batch-to-Batch Efficacy of Corneal Scar Inhibition by Human CSSC

In a mouse model of anterior stromal injury caused by mechanical ablation, human CSSC (50 × 10^3^ cells) loaded in fibrin gel was topically applied to fresh corneal wounds. At day 14 post-treatment, the corneas were examined for clarity and scar formation. Our results summarizing the treatment using 24 primary CSSC cultures derived from different donor corneas showed a correlation of treatment outcomes with the CSSC features (Figure 3).

Primary cultures with cells exhibiting small size and clonal expansion (n = 8) exhibited a high preventive effect for scar formation (62.5% resulting in scale 0, with minimal opacities; 25% in scale 1, with mild opacities, and 12.5% in scale 2, with moderate opacities,). None of these cells resulted in intense scarring.Cultures having mixed cell morphologies of small and slender shapes (n = 14) were incapable of completely preventing scar formation (0% for scale 0). Instead, they produced slight to moderate scarring (42.8% for both scale 1 and 2), and occasionally showed intense scarring (14.3% for scale 2).Cultures with a dominant appearance of bipolar and slender-shaped cells (n = 2) yielded intense scarring (100% for scale 3).

### 3.3. Cell Dosage-Dependent Scar Inhibition by Both Human CSSC and CSK Treatments on Mouse Stromal Injury

With reference to (1) Basu et al. [13] reporting the topical treatment with 50 × 10^3^ CSSC in fibrin gel, and (2) Du et al. [16] using a stromal injection of 50 × 10^3^ CSSC, the stem cell treatments prevented corneal scarring in mouse corneas after injury and due to congenital lumican knockout, respectively. Here, we tested 3 different doses of primary human CSSC (30, 50, and 70 × 10^3^ cells per treatment) for their anti-scarring effects after topically applied on fresh corneal wounds of Swiss Webster mice (Charles River Lab, Malvern, PA, USA). The experiment was repeated with 3 CSSC cultures from different donor corneas. At day 14 post-treatment, overall, the corneas displayed a scar inhibitory effect (Figure 4A). The percentage of scar area over the entire corneal surface was dose-dependently reduced. Compared to the injured controls, treatment with 50 × 10^3^ cells and above significantly reduced the extent of corneal scarring (Figure 4B). when analyzed using ASOCT images, the mean corneal thickness (CCT) was significantly greater in the untreated wound corneas than in the naïve controls (the thickness increased by 62% at week 1, and 94% at week 2 after injury) (*p* < 0.05, Mann–Whitney U test); CSSC treatments notably reducing corneal thickening, particularly at week 2 post-treatment, irrespective of cell doses (Figure 4C). The expression of fibrotic gene markers (*COL3a1*, *αSMA*, *FN*, *TNC*) were also significantly suppressed in corneas treated with 50 and 70 × 10^3^ CSSC, respectively (Figure 4D).

Alternatively, q-CSK were injected intrastromally to mouse corneas at 1-week post-injury, after the healing of the corneal epithelium. Due to the lack of mouse data, we calculated that the mouse cornea contains 17 × 10^3^ CSK, based on the mean corneal diameter of ~2.5 mm and thickness of 0.12 mm [27], and with reference to the mean keratocyte density of 20 × 10^3^/mm^3^ in human corneas [28]. We tested 3 different doses of q-CSK (10, 20, and 30 × 10^3^ cells) for stromal injection in 0.5 μL volume. The experiment was repeated with 3 different CSK cultures from the same donor corneas generating CSSC. At day 14 post-treatment, mouse corneas injected with 10 × 10^3^ CSK showed moderate scarring, although the scarring was relatively less than in the injury-only and PBS-injected controls (Figure 5A). Corneas injected with 20 and 30 × 10^3^ q-CSK showed scar inhibition (a lower percentage of scar area), particularly at week 2 post-treatment (*p* < 0.05, Mann–Whitney U test) (Figure 5B). Various fibrotic gene expression was also significantly downregulated in injured corneas treated by a 20 and 30 × 10^3^ q-CSK injection (Figure 5D).

### 3.4. Fibrotic Phenotypes after CSSC vs. q-CSK Treatments of Mouse Stromal Wounds

Immunofluorescence showed that mouse Col3a1, αSMA, FN, and TNC were strongly detected in the corneal stroma at 2 weeks post-injury (Figure 6), indicating the stromal fibrosis and scar formation, as shown in the isolated corneal samples (Figure 4 and Figure 5). Most of the gene expression was greatly reduced after CSSC treatment (topical application of 50 × 10^3^ cells), except for FN, which showed mainly sub-epithelial, but weak stromal expression. On the other hand, the intrastromal injection of q-CSK (20 × 10^3^ cells) substantially suppressed the overall expression of fibrotic genes, when compared to the wound controls (Figure 6). Some minor αSMA positive signals could correspond to the minor haze appearing on the isolated corneas in Figure 5A. All antibodies used were rabbit raised to react against mouse antigens, and combined with sections without primary antibodies, we confirmed there was negligible non-specific staining.

### 3.5. Combined CSSC and q-CSK Treatments of Mouse Corneal Stromal Wounds

Mouse corneas with fresh stromal injuries and open epithelial wounds (n = 15) were treated with human CSSC (50 × 10^3^ cells) loaded in fibrin gel. After 7 days of epithelial healing, the corneas were treated or not with an intrastromal injection of q-CSK (20 × 10^3^ cells) (each group n = 5). The experiment was repeated with 3 pairs of CSSC and CSK derived from donor corneas. After 2 weeks, the isolated corneas showed that a single CSSC treatment substantially reduced corneal scar formation (Figure 7A), when compared to wound controls, and this result was similar to our previous findings. The percentages of the scar area were significantly reduced (26.1 ± 3.4%, 18.9 ± 4.3%, and 29.1 ± 7.2%) after treatment with 3 different CSSC batches, in contrast to 66.1± 8.1% for wound controls; *p* < 0.05, Mann–Whitney U test) (Figure 7B). In the group with additional CSK injections, the remaining opacities were further inhibited and the corneal clarity was better, when compared to single CSSC-treated corneas. A further reduction in the scar area percentages (10.2 ± 4%, 8.9 ± 2.2%, and 10.7 ± 4.1%) resulted. Compared to the wound controls, corneas with combined cell treatment had significantly inhibited corneal scarring (*p* < 0.01) (Figure 7B).

Fibrotic gene expression analyzed by qPCR substantiated the scar inhibition results. The expression of *COL3a1*, *αSMA*, *FN*, and *TNC* in CSSC-only treated corneas were significantly suppressed, when compared to wound controls (*p* < 0.05, one-way ANOVA) (Figure 7C). The treatment with additional CSK injections further reduced these gene expressions, reaching levels close to those of the naïve corneas. Significant downregulation of *COL3a1* and *αSMA* was observed, when compared to the single CSSC-treatment (*p* < 0.05).

Corneal transparency is determined by the highly regulated fibrillar pattern of the collagenous stromal ECM. Thus, it is important to assess the collagen fibril arrangement of the cell-treated corneas compared to the wound controls. Under transmission electron microscopy, the stromal scar region of the wounded corneas lacked the characteristic lamellae organization, which was readily detectable in the naïve stroma. In the native stromal ECM, collagen fibrils were organized into lamellae containing small, uniform, and regularly aligned fibrils (Figure 8A). In contrast, the scar tissue consisted of irregularly-aligned collagen fibrils and amorphous deposits, and the interfibrillar spacing was inconsistent. A 360° fibril distribution profile revealed an asymmetrical pattern, which was different from the consistent and balanced configuration seen in the naïve stroma (Figure 8B). This ECM fibrillar pattern derived from the scar tissue could lead to the disturbance in the light passage. In corneas treated with combined CSSC and CSK, the collagen fibril organization was restored and closely resembled the native tissue. Higher magnification of the fibrils showed less irregular spacing, and the distribution profile was similar to that in the native stroma (Figure 8C). In terms of the interfibrillar distance between unblocked fibrils, the combined cell-treated corneas were also similar to the native corneas (Figure 8D). Although the mean interfibrillar distances in the wounded stromal tissue did not show any significant difference from the normal corneas, they spanned a much wider range, indicating their irregularity. The wound tissue exhibited interfibrillar distances from 37 to 128 nm (n = 15 fields), whereas the native tissues had a narrow range of 54 to 72 nm (n = 15 fields). Cell-treated corneas also had a more restricted range, from 60 to 87 nm (n = 15 fields).

## 4. Discussion

In this study, we showed that primary human CSSC and CSK (both activated and quiescent) possessed different features, including differential stem cell and stromal marker expression, collagen secretion, and anti-inflammatory potency. While a clinical trial using CSSC treatment for corneal scar management is underway, the in-batch as well as the batch-to-batch variations of CSSC affecting the treatment outcome should not be overlooked. Here, we demonstrated a dose-related response after topical CSSC treatment for preventing corneal opacities due to stromal ECM remodeling after injury. A similar cell dose-related treatment outcome was also observed for CSK delivered via intrastromal injection. Since CSSC and CSK were administered via different routes, our study did not directly compare their treatment efficacies. Using optimal doses, we found that the CSSC treatment effect on suppressing corneal opacities was augmented by an additional CSK injection, resulting in a recovery of corneal clarity. These in vivo effects were supported by the downregulated expression of stromal fibrosis genes and the restoration of stromal fibrillar organization and regularity. Our results thus indicated that a combined CSSC and CSK treatment could achieve a higher clinical efficacy of a scarless corneal stromal healing and could restore corneal transparency, when compared to a single CSSC treatment.

In our mouse model of corneal epithelial-stromal injury caused by the mechanical debridement using Algerbrush burring, the wound healing response involves a cascade of events leading to stromal ECM remodeling (the formation of opacities and scarring), which affects the normal stromal structure and functions [2,29]. An early inflammatory reaction due to the release of cytokines (such as interleukin-1, IL-1, and TNFα) from the damaged epithelium and Bowman’s layer [30], not only causes the keratocytes at the wound site to undergo apoptosis, but also activates the viable keratocytes in the vicinity to transit into the repair-type stromal fibroblasts (SF) [31,32]. SF further produce pro-inflammatory cytokines (e.g., MCP-1), attracting inflammatory cells, such as neutrophils and platelets, to enter into the stroma from peripheral vasculatures [33]. This feedback loop results in further SF generation and repopulation [34]. SF also express fibronectin receptors, produce and deposit repair-type ECM proteins (e.g., fibronectin and SPARC), and collagenases to mediate tissue remodeling [2]. In addition, under TGFβ1/2 signaling from the damaged epithelium and apoptotic CSK, SF further transform into αSMA-positive myofibroblasts. The presence of myofibroblasts, via expressing fibronectin receptors (α5β1 and αvβ3 integrins), promotes the assembly of fibronectin fibrils, and this interaction exerts mechanical forces for wound matrix contraction [35]. SF and myofibroblasts excessively produce and deposit abnormal ECM components (e.g., collagens, biglycan, SPARC) in a disorganized manner inside the corneal stroma, hence compromising the corneal transparency. Myofibroblasts may also compromise a heterogenous cell mix; some have been shown to be CD11-positive, suggesting that they could originate from bone marrow-derived fibrocytes [36]. Myofibroblasts were also reported to be derived from non-myelinating Schwann cells upon ERK activation [37].

After injury, tissue inflammatory responses caused by the early neutrophil infiltration are linked to downstream fibrosis and scar formation [38,39]. Though neutrophil recruitment possesses a series of antibacterial and antiseptic properties, such as scavenging tissue debris, different studies have found that the neutrophil infiltration is positively correlated with the extent of tissue fibrogenesis [40,41]. In addition, neutrophils secret toxic mediators, such as reactive oxygen species and reactive nitrogen species, can lead directly to tissue injury, or even worse, can induce another wave of chemokines that form a positive feedback loop for tissue damage [42]. Hence, suitable treatment to suppress an early inflammatory response is beneficial to control the subsequent fibrotic initiation and progression. One may argue that non-steroidal anti-inflammatory drugs (NSAIDs) have been widely prescribed to reduce inflammation, particularly after surgery. NSAID treatment is effective to suppress cyclooxygenases (COXs), a key mediator of inflammatory pathways [43]; however, NASAIDs have been shown to be associated with corneal thinning and perforation. This could be due to the reduced levels of COX product 12(S)-hydroxyheptaseca-5Z,8E,10E-trienoic acid (12-HHT) after COX suppression [44]. The endogenous ligand for leukotriene B4 receptor 2 is 12-HHT, which is important for tissue homeostasis, including corneal tissues [45]. Hence, NSAID eye drops that inhibit the production of 12-HHT delays corneal wound healing [43].

Human CSSC, owing to their shared properties with MSC, play an important role in suppressing inflammation, especially when applied to an acute wound situation [46,47]. This was demonstrated by the reduced neutrophil counts in mouse corneas after wounding followed by CSSC engraftment [19]. This effect appears to be mediated via the TSG-6 pathway by interacting with the chemokine CXCL8 to suppress neutrophil migration. In addition, Gu et al. showed that MSC treatments decreased the expression of COX-2 and NFκB, thereby reducing the production of pro-inflammatory cytokines [48]. This effect was related to the ability of MSC to inhibit the phosphorylation of ERK and the p38/MAPK enzymes and pathway, which modulated the inflammatory response. Whether human CSSC suppress COX expression in corneal tissues remains to be elucidated. On the other hand, CSK were sensitive to pro-inflammatory cytokines, including IL-1 and TNFα, which caused the cells to become non-functional and apoptotic [49,50]. In our study, quiescent CSK had a relatively less suppressive effect on mouse macrophages treated with RANKL and ConA for undergoing chronic pro-inflammatory osteoclastogenesis than CSSC, indicating that this anti-inflammatory function is more relevant to CSSC. As a whole, CSSC treatment has the ability to reduce corneal tissue inflammation at early time points after injury, and to subsequently block fibrosis. However, whether these effects have sufficient power regarding stromal tissue regeneration is highly related to the batch quality of CSSC, including the cell homogeneity in cultures (see discussion in next section), survival of CSSC in tissue wound condition, and their capacity to differentiate into keratocytes. In an injured tissue environment with the presence of multiple interfering factors, such as pro-inflammatory cytokines and growth factors—TGFβ, PDGF, FGF, etc.,—the normal differentiation of CSSC to CSK could be affected, and this could pose a risk to the stromal recovery and visual restoration.

Stromal ECM restoration requires a good source of collagen. Inside the corneal stromal matrix, Col1 is a predominant collagen type, supplemented with small amounts of types V, VI, XII, XIII, XIV, and XXIV [51]. When compared to quiescent CSK, CSSC produced and deposited much lower levels of Col1A1 protein (almost 1/10 the amount of q-CSK), indicating that CSSC, before fully differentiating into mature keratocytes, are inefficient in achieving stromal remodeling towards a native direction. Such keratocyte differentiation capacity of CSSC is highly related to the batch quality and the stability of cells that could be influenced by the presence of serum factors and cytokines in the wound site. These factors are thus important to affect the outcomes of CSSC therapy. After understanding this treatment limitation, we introduced an additional q-CSK injection to the corneal stroma after stabilization by CSSC treatment (i.e., suppressed inflammation and fibrosis). The injected CSK directly deposited collagens and KSPG to assist the native stromal ECM remodeling. The treated corneas exhibited a higher level of clarity and proper collagen fibril configuration, similar to naïve corneas. The expression of various fibrosis markers was further downregulated, when compared to single CSSC treatment, in which the levels had been significantly reduced with regards to the wound controls. These results support the idea that early CSSC treatment exerts anti-inflammatory activity, primarily by reducing the infiltration of neutrophils, leading to different degree of delayed fibrosis, and inhibiting the abnormal ECM remodeling. A subsequent CSK injection provides a collagen matrix and proteoglycans for a native-like stromal ECM recovery, improving corneal transparency.

Human CSSC are firstly isolated from the anterior limbal stroma where other cell types are present in close proximity, including melanocytes, nerve, and vascular endothelial cells [52]. From cell culture studies, primary CSSC exhibit clonal growth, exist as a “side population” to efflux Hoechst 33,342 DNA-binding dye in fluorescence-activated cell sorting (FACS), and show shared properties with undifferentiated MSC, particularly bone marrow MSC, expressing various mesenchymal lineage markers (CD73, CD105) and stem cell markers, such as ABCG2 [13,22]. However, definitive markers for CSSC are yet to be identified. Their differentiation to keratocytes has been demonstrated under an in vitro reduced mitogen condition supplemented with ascorbic acid and TGFβ3, resulting in upregulated keratocyte-specific KSPG expression, e.g., keratocan [22]. The stromal injection of human CSSC into normal mouse corneas and thin corneas of lumican null mice showed an accumulation of human keratocan, indicating cell differentiation to keratocyte lineage in vivo [16]. Similar to in vitro studies, the possible autocrine action of TGFβ3 produced by CSSC could drive the keratocyte differentiation and stromal ECM remodeling, improving corneal clarity and thickness [18]. Treatment with CSSC with TGFβ3 knockdown showed a loss of such effects. However, when applied to the wound, it remains to be elucidated whether CSSC can be maintained or undergo preferred differentiation to keratocytes, since there are higher levels of pro-fibrotic TGFβ1 and β2, rather than TGFβ3, present in vivo [53]. Moreover, the manner in which the proliferating CSSC avoids over-populating the wound site needs to be studied. Hence, our combined cell treatment approach applied CSSC at an early time point post-injury to control tissue inflammation and delay the onset of fibrosis, aiming to stabilize the wound condition. The next injection of q-CSK augmented the CSSC effect by preventing abnormal ECM remodeling (anti-scarring). Characterized by unique markers, including keratocan and its biosynthesizing enzymes (B3GnT7 and CHST6), as well as ALDH3A1, the quiescent differentiated CSK provided stromal collagen for ECM recovery. Our previous proof-of-concept study in a rat model, with corneal haze formed due to laser injury, illustrated that the injected CSK behaved like the native keratocytes, synthesizing and depositing human KSPG and Col1 [14]. This effect is well substantiated by our transmission EM study, showing that the collagen fibrils and their interfibrillar pattern became more regularly aligned than those in the scar tissue. This balanced configuration of collagen fibrils is less likely to block or deviate light passage through the stromal tissue. In addition, our previous time-lapse study showed that the injected CSK stayed inside the corneal stroma, with a half-life of about 3 weeks, and 20% of the injected cells remained after 6 weeks [14]. This indicates that the injected CSK could assist in long-term stromal tissue remodeling.

Promising results are shown regarding cell-based therapy using sequential CSSC and CSK treatments for corneal scar prevention immediately after injury. The early CSSC intervention to suppress tissue inflammation and delay fibrosis, followed by quiescent CSK delivery to provide collagens and KSPG for corneal stromal ECM remodeling, is a novel concept to restore native-like tissue regeneration. However, individuals with corneal wounds have a very limited opportunity to obtain medical care in the brief period of time during which inflammation manifests in the injured tissue. It is therefore important to explore clinically relevant molecules similar to the mechanisms by which CSSC control inflammatory reactions. Understanding and utilizing this feature, combined with the CSK providing necessary stromal proteins, will allow for the treatment of individuals with corneal injury and subsequent scarring.

## Figures and Tables

**Figure 1 ijms-23-06980-f001:**
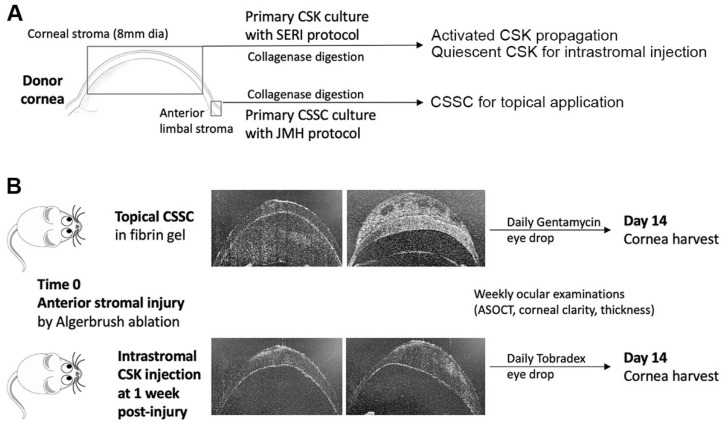
Schematic diagram of tissue sources of CSSC and CSK, and cell treatment modalities for mouse model of corneal injury. (**A**) From healthy donor cornea, the central stroma was harvested for CSK isolation and primary culture. Activated CSK were propagated to passage 3 using SERI protocol, and cells at passage 4 were changed to the serum-free ERI condition to generate quiescent CSK (q-CSK) for characterization and treatment use. The anterior limbal stroma from the same donor cornea was collected for CSSC isolation. The primary culture of CSSC at passage 3 was used for characterization and treatment. (**B**) In a mouse model of epithelial-stromal injury created by mechanical ablation using Algerbrush, CSSC in fibrin gel was loaded onto the corneal wound surface immediately after injury. Quiescent CSK was intrastromally injected at 1-week post-injury. All corneas were harvested at day 14 post-treatment.

**Figure 2 ijms-23-06980-f002:**
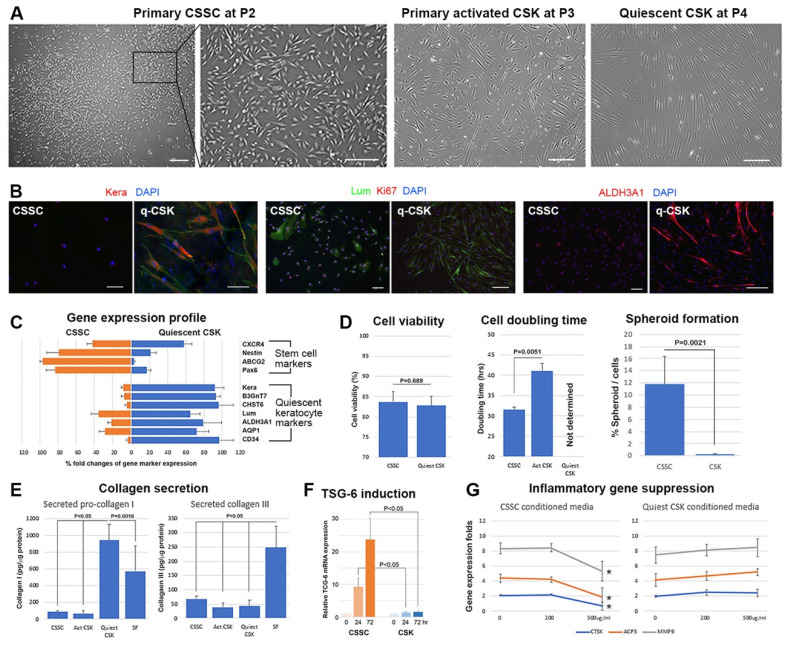
Characterization of CSSC and CSK. (**A**) Phase contrast images of the primary cell culture; scale bars: 100 μm. (**B**) Immunofluorescence of keratocan (Kera), lumican (Lum), aldehyde dehydrogenase 3A1 (ALDH3A1), and Ki67 for CSSC and CSK; scale bars: 50 μm. (**C**) Gene expression profile of stem cell and corneal stromal markers by qPCR analysis. (**D**) Culture characterization: viability, doubling time, and spheroid formation (one-way ANOVA test). (**E**) Expression of collagen 1 and III in culture media (Mann–Whitney U test). (**F**) Expression of *TSG-6* in cell cultures treated with TNFα. Time-dependent upregulation of TSG-6 in CSSC cultures, but not in CSK culture (one-way ANOVA test). (**G**) Osteoclast gene expression (*ACP5*, *MMP9* and *CTSK*) of mouse RAW cells induced by RANKL and ConA treatment and a dose-dependent suppression by CSSC conditional media (CM) concentrates (* *p* < 0.05, compared to non-CM treated control). No inhibition effect was observed for CSK CM concentrates.

**Figure 3 ijms-23-06980-f003:**
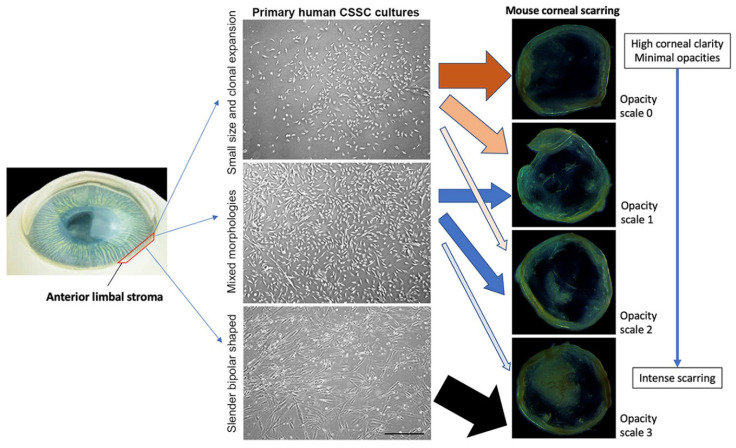
Batch variation of CSSC cultures influencing the treatment outcomes. Primary human CSSC exhibited heterogenous features in the culture, from the cultures with generally small size and clonal proliferation, to mixed morphologies, with slender-shaped cells combined with small-sized cells. While the majority of small-sized CSSC, the cultures could suppress corneal scar formation (brown arrows); other CSSC batches with mixed morphologies provided moderate effects on scar inhibition (blue arrows). Using batches with predominantly slender-shaped cells, the anti-scarring effect was minimal (black arrow). The cultures of these poor-quality cells were usually terminated and discarded; scale bar: 100 μm.

**Figure 4 ijms-23-06980-f004:**
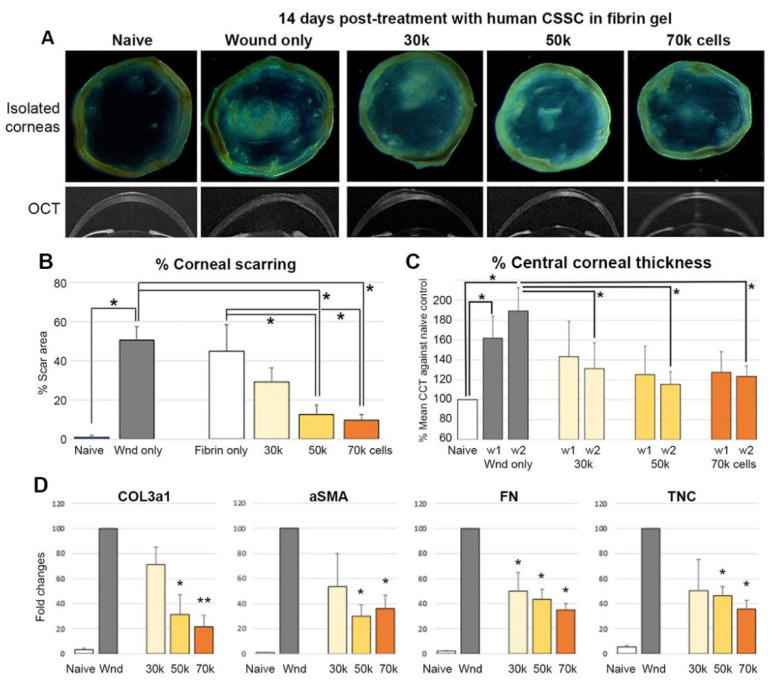
Treatment of CSSC in a mouse model of epithelial-stromal injury. (**A**) Isolated cornea and representative OCT images at week 2 post-injury, and cell treatment with different doses (30–70 × 10^3^ cells) in fibrin gel. (**B**) A cell dose-dependent reduction in the percentage of the scar area (* *p* < 0.05). (**C**) The percentage of the central corneal thickness (* *p* < 0.05). (**D**) Fibrosis gene expression by qPCR (* *p* < 0.05, ** *p* < 0.01, compared to wound control, Mann–Whitney U test).

**Figure 5 ijms-23-06980-f005:**
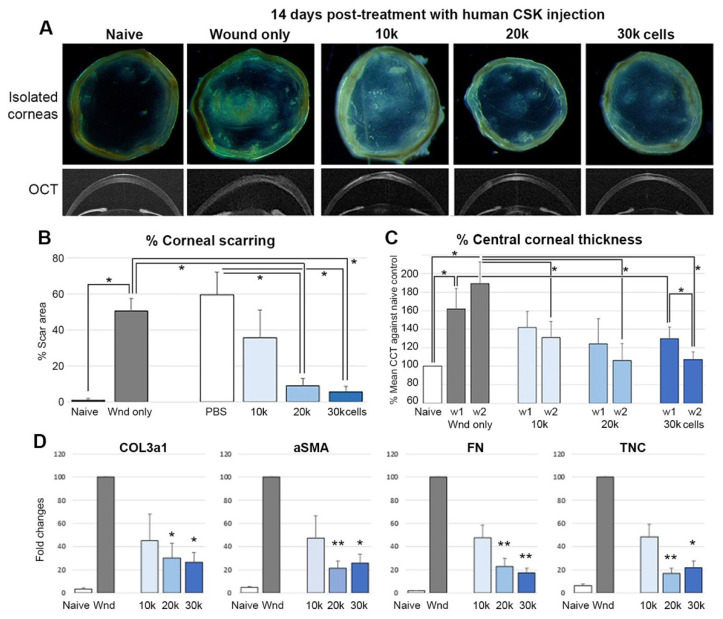
Treatment of CSK via stromal injection in a mouse model of epithelial-stromal injury. (**A**) Isolated cornea and representative OCT images at week 2 post-cell treatment with different doses (10-30 × 10^3^ cells). (**B**) A cell dose-dependent reduction in the percentage of the scar area (* *p* < 0.05). (**C**) The percentage of the central corneal thickness (* *p* < 0.05). (**D**) Fibrosis gene expression by qPCR (* *p* < 0.05, ** *p* < 0.01, compared to wound control, Mann–Whitney U test).

**Figure 6 ijms-23-06980-f006:**
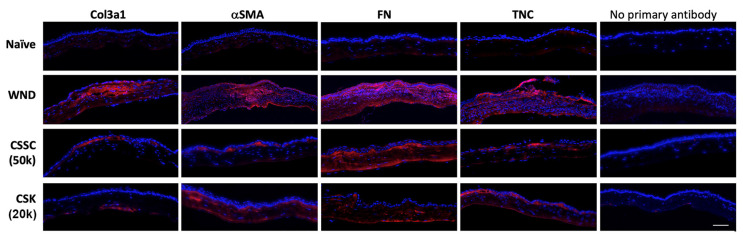
Immunostaining of fibrosis markers in mouse corneal sections after cell treatments. The expression of Col3a1, αSMA, fibronectin (FN), and tenascin C (TNC) was examined in mouse corneas (naïve and wound controls), compared to corneas treated with 50 × 10^3^ CSSC or 20 × 10^3^ q-CSK for 2 weeks; scale bar: 50 μm.

**Figure 7 ijms-23-06980-f007:**
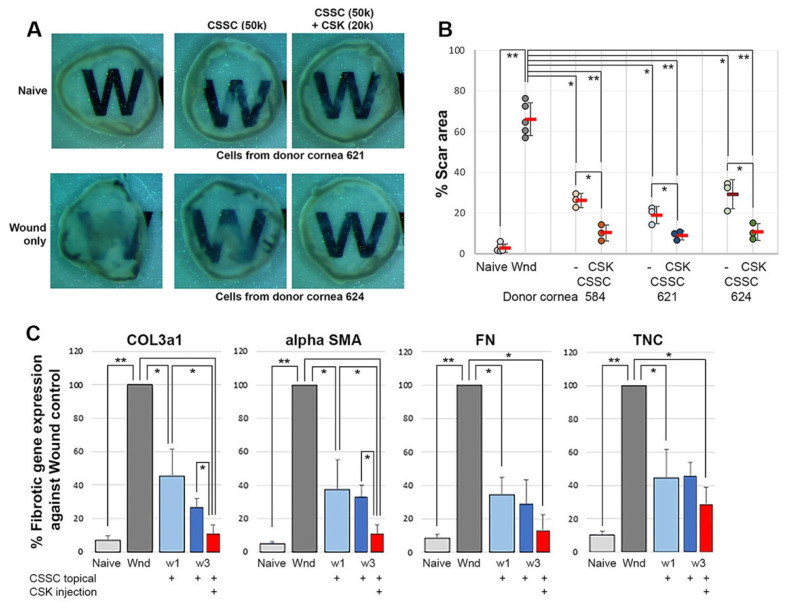
Combined CSSC and CSK treatments on mouse corneas after epithelial-stromal injury. (**A**) Isolated cornea images. Corneas after combined cell treatment were more clear than single CSSC-treated corneas, showing minor haze. Wounded corneas had extensive scarring. (**B**) Further reduction in the percentage of the scar area after combined cell treatment, when compared to single CSSC treatment. The experiment was performed using 3 different pairs of CSSC and CSK from the same donor corneas (* *p* < 0.05, ** *p* < 0.01). (**C**) Fibrosis gene expression by qPCR (* *p* < 0.05, ** *p* < 0.01, compared to wound control, Mann–Whitney U test).

**Figure 8 ijms-23-06980-f008:**
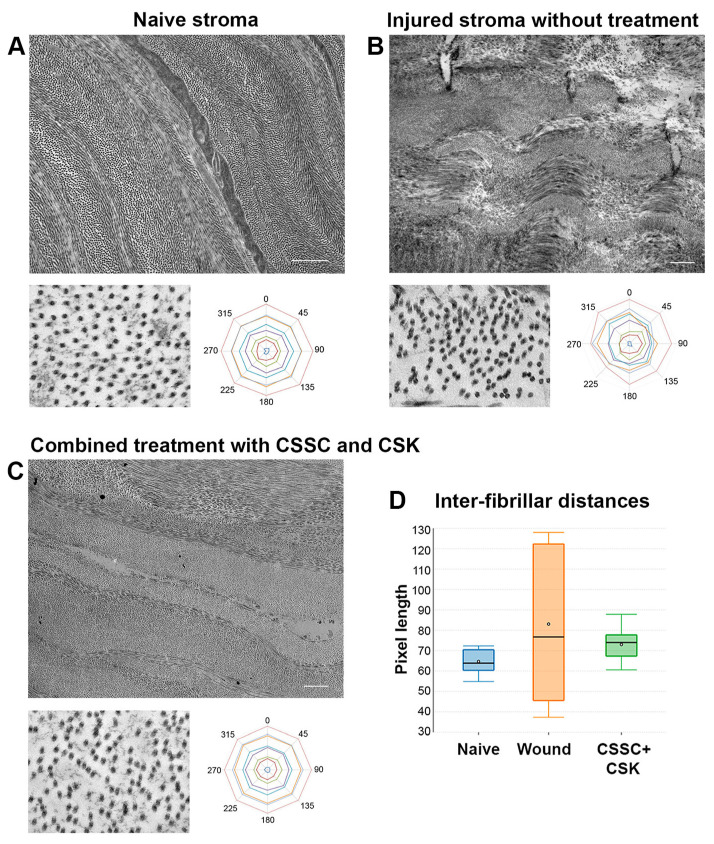
Transmission electron microscopy analysis of mouse corneas after combined CSSC and CSK treatment. (**A**) Naïve stroma with regularly aligned collagen fibrils and balanced fibrillar configuration. (**B**) Scarred stroma with variable collagen packing and irregular fibril distribution. (**C**) Corneas after combined cell treatment had compact collagen organization with a more consistent fibril arrangement. (**D**) Scarred stroma showed a wider range of interfibrillar distances than naïve stroma. Cell-treated stroma showed a similar narrow range of distances; scale bars: 2 μm.

## Data Availability

All data are included in the text and Appendix A.

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
