# Peer review of "Combined Therapy Using Human Corneal Stromal Stem Cells and Quiescent Keratocytes to Prevent Corneal Scarring after Injury"

_ijms, 2022, doi:10.3390/ijms23136980_

Round 1
Reviewer 1 Report
An interesting and well designed study of the effect of administering two different formulations/phenotype of corneal/limbal stromal cells to damaged corneas. A significant contribution to the field.
1. In the second to last line of the introduction, it is more accurate to use the phrase "immuno-regulatory nature" rather than "mesenchymal nature" to avoid confusion with broader use of the term "mesenchymal" other than for MSC.
2. While published previously, I think it would be useful for readers of this manuscript to have complete details for the composition of growth media used (SERI, ERI and JM-H).
3. The phase contrast images displayed in Figure 2 should be presented at full page width in order to see morphological differences more clearly.
4. The final graph in Figure 2 has a typo. "Quest" should be Quiesc."
5. "Algerbrush burring" should be changed to "mechanical debridement".
6. Phase contrast images for Figure 3 should also be presented at twice the present size in order to see details.
7. Individual parts for Figures 2, 4, 5 and 7 require labels (i.e., A, B, C etc) to match legends and help direct reader to each piece of data.
8. Need to check for consistent use of a space before units of measurement (%, nm etc).
9. Please check scale bars used for Figure 8. Should this be "µm" rather than "mm"?
END COMMENTS
Author Response
An interesting and well-designed study of the effect of administering two different formulations/phenotype of corneal/limbal stromal cells to damaged corneas. A significant contribution to the field.
Reply – We thank the reviewer for the great effort and constructive comments.
- In the second to last line of the introduction, it is more accurate to use the phrase "immuno-regulatory nature" rather than "mesenchymal nature" to avoid confusion with broader use of the term "mesenchymal" other than for MSC.
Reply – Thank you for the suggestion. We changed “mesenchymal” to “immuno-regulatory” in the abstract and introduction (page 2, third paragraph).
- While published previously, I think it would be useful for readers of this manuscript to have complete details for the composition of growth media used (SERI, ERI and JM-H).
Reply – Thank you for the suggestion. We listed the media composition in M&M (page 3).
- The phase contrast images displayed in Figure 2 should be presented at full page width in order to see morphological differences more clearly.
Reply – Thank you for the suggestion. We revised the figure and enlarged the cell images for a clearer presentation.
- The final graph in Figure 2 has a typo. "Quest" should be Quiesc."
Reply – Thank you. It was corrected.
- "Algerbrush burring" should be changed to "mechanical debridement".
Reply – Thank you. It was corrected in Fig. 1 legend, Results on page 9, and Discussion on page 15.
- Phase contrast images for Figure 3 should also be presented at twice the present size in order to see details.
Reply – Thank you for the suggestion. The cell images were enlarged.
- Individual parts for Figures 2, 4, 5 and 7 require labels (i.e., A, B, C etc) to match legends and help direct reader to each piece of data.
Reply – Thank you. Sub-figure labels were added.
- Need to check for consistent use of a space before units of measurement (%, nm etc).
Reply – Thank you. All spacings were checked.
- Please check scale bars used for Figure 8. Should this be "µm" rather than "mm"?
Reply – Thank you. The symbol font was revised.
Reviewer 2 Report
The present study reports the treatment of corneal opacity by combining the topical immediate application of CSSC and one week later injection of CSK. There is the question whether the recovery is partial supported by the fact that the limbus is still intact. The mouse model did not have a limbectomy. the authors need to elaborate on if corneal epithelium debridement included limbus removal.
Why CSSC were applied first and why could net the CSSC be injected later on instead of injecting CSK?
How is combination of CSSC + CSK addressing the batch-to-batch issue. These are not autologous cells, there still be the issue of donor batch-to-batch issue. Did the injection of CSK cause a certain level of stromal opacification. Were the cells pre-labelled to follow their distribution once they were administered to the eye? did the cells stayed at the application site?
The figures do not include the A,B and tec. labels.
Overall, the study is highly interesting to read and highly impactful.
Author Response
- The present study reports the treatment of corneal opacity by combining the topical immediate application of CSSC and one week later injection of CSK. There is the question whether the recovery is partial supported by the fact that the limbus is still intact. The mouse model did not have a limbectomy. The authors need to elaborate on if corneal epithelium debridement included limbus removal.
Reply – Thank you for reviewing our manuscript and the constructive comments. In this mouse model, we aimed to create anterior stromal scarring by mechanically damaging the corneal epithelium and Bowman’s membrane in the central corneal region. The limbus was kept intact and only the central epithelium (2 mm diameter out of the entire corneal diameter of ~2.6 mm) was removed by Algerbrush ablation. The epithelium was allowed to heal by the endogenous transit-amplifying cells and limbal epithelial stem cells and this usually took 3-5 days to complete. In M&M (on page 5), we have indicated that the epithelial removal did not affect the limbus. In addition, the wound control corneas without cell treatments exhibited intense corneal scarring, indicating that the presence of intact limbus did not affect the stromal scarring or modulate the treatment effects.
- Why CSSC were applied first and why could net the CSSC be injected later on instead of injecting CSK?
Reply – Thank you. This is exactly our hypothesis that the immuno-regulatory nature of CSSC can stabilize the tissue wound by controlling inflammation which usually occurs at an early time after injury, thereby delaying the onset of fibrosis. A subsequent intrastromal injection of CSK, which synthesize and deposit stromal-specific collagens (particularly Col 1) and KSPG, improve the stromal remodeling, and restore the collagen fibrillar organization and regularity, resulting in a better recovery of corneal clarity. Whether this healing effect can be achieved by CSSC injection is questionable and needs to be verified in future experiments. But we know that CSSC, per se, are not specialized to produce stromal collagens and KSPG as compared to the mature and quiescent CSK. They might need to differentiate towards more keratocyte-like in order to possess such capability. However, unlike normal embryonic development with a regulated keratocyte-lineage differentiation, the injured tissue environment with the presence of multiple interfering factors (like pro-inflammatory cytokines and growth factors - TGFb, PDGF, FGF, etc) could affect CSSC differentiation and generate cells other than the keratocytes. This could pose a major risk to stromal recovery and visual restoration. This point has been included in our discussion on page 16.
- How is combination of CSSC + CSK addressing the batch-to-batch issue. These are not autologous cells, there still be the issue of donor batch-to-batch issue. Did the injection of CSK cause a certain level of stromal opacification. Were the cells pre-labelled to follow their distribution once they were administered to the eye? did the cells stayed at the application site?
Reply – Both CSSC and q-CSK from the same donor corneas were characterized prior to using them in the animal study. We selected the combinations according to their gene marker expression profile as shown in Fig. 2C. Primary q-CSK showed the unique expression of keratocan, and its synthesizing enzymes (B3GnT7 and CHST6), as well as CD34 to indicate quiescence. On the other hand, CSSC expressed stem cell markers ABCG2, nestin, and Pax6, as well as MSC markers CD73, 90, and 105. However, the lack of a definitive marker for CSSC could limit the accuracy. This point has been mentioned in the Discussion on page 16 last paragraph. In the present study, CSK injection exhibited a dose-related suppression of corneal scarring and haze formation after anterior stromal injury. As shown in our previous study, CSK injection into normal rat corneas did not cause corneal opacities and most corneas were optically clear (Yam et al. 2018 Invest Ophthalmol Vis Sci). In the same study, we also used Spectralis HRA-OCT to examine the time-lapse cell intensity and distribution changes after the injection of ION-labeled CSK. Our result showed a time-dependent reduction of CSK signal intensity with approximately 50% cells remaining after 3 weeks and 20% cells after 6 weeks post-injection. This point has been added in the Discussion on page 17. The distribution of CSK primarily stayed in the region where the injection bleb was formed. Whether they will migrate through stromal layers will be of interest to be studied under transmission EM.
- The figures do not include the A,B and tec. labels.
Reply – Thank you. Sub-figure labels were added.
Overall, the study is highly interesting to read and highly impactful.
Reply – Thank you.